# Validation of maternal recall of number of antenatal care visits attended in rural Southern Nepal: a longitudinal cohort study

Xinyu Xie ,[1] Melinda K Munos ,[1] Tsering P Lama,[1] Emily Bryce ,[1] Subarna K Khatry,[1,2] Steven C LeClerq,[1,2] Joanne Katz[1]

[1]Department of International Health, Johns Hopkins University Bloomberg School of Public Health, Baltimore, Maryland, USA
[2]Nepal Nutrition Intervention Project Sarlahi, Lalitpur, Nepal

**Correspondence to**
Xinyu Xie;
evaxie.shchn@gmail.com

## ABSTRACT

**Objectives** This study aimed to examine the validity of maternal recall of total number of antenatal care (ANC) visits during pregnancy and factors associated with the accuracy of maternal recall.

**Design** This was a longitudinal cohort study conducted from December 2018 through November 2020.

**Setting** Five government health posts in the Sarlahi district of Southern Nepal.

**Participants** 402 pregnant women between ages 15 and 49 who presented for their first ANC visit at the study health posts.

**Main outcomes** The observed number of ANC visits (gold standard) and the reported number of ANC visits at the postpartum interview (maternal recall).

**Results** On average, women in the study who had a live birth attended 4.7 ANC visits. About 65% of them attended four or more ANC visits during pregnancy as recommended by the Nepal government, and 38.3% of maternal report matched the categorical ANC visits as observed by the gold standard. The individual validity was poor to moderate, with the highest area under the receiver operating characteristic curve (AUC) being 0.69 (95% CI: 0.65 to 0.74) in the 1–3 visits group. Population-level bias (as distinct from individual-level bias) was observed in the 1–3 visits and 4 visits groups, where 1–3 visits were under-reported (inflation factor (IF): 0.69) and 4 ANC visits were highly over-reported (IF: 2.12). The binary indicator ANC4+ (1–3 visits vs 4+ visits) showed better population-level validity (AUC: 0.69; IF: 1.17) compared with the categorical indicators (1–3 visits, 4 visits, 5–6 visits and more than 6 visits). Report accuracy was not associated with maternal characteristics but was related to ANC frequency. Women who attended more ANC visits were less likely to correctly report their total number of visits.

**Conclusion** Maternal report of number of ANC visits during pregnancy may not be a valid indicator for measuring ANC coverage. Improvements are needed to measure the frequency of ANC visits.

## STRENGTHS AND LIMITATIONS OF THIS STUDY

⇒ The gold standard was established using direct observation by trained field workers, thus eliminating the risk of recall bias.
⇒ The study observers were all trained to reach a standard level of validity before working at the study sites, which provides a more objective and reliable source for verification than secondary databases.
⇒ The study had an appropriate length of recall period comparing to other validation studies who use recall periods of less than 6 months or even exit interviews to validate maternal report.
⇒ The study only considered women who presented for their first antenatal care (ANC) at public health posts for the feasibility of data collection.
⇒ Women who visited facilities other than the study health posts were not observed but were asked to recall how many other ANC visits they attended.

global neonatal mortality rate of 18 deaths per 1000 live births in 2021.[1 2] Maternal and neonatal mortality remains an issue that differentially impacts developed and developing countries. According to the WHO, 94% of all maternal deaths in 2017 occurred in low-income and middle-income countries, with 86% taking place in sub-Saharan Africa and South Asia.[2] Sub-Saharan Africa and South Asia also have the highest neonatal mortality rate among all regions (27 and 23 deaths per 1000 live births, respectively, in 2021).[1]

Antenatal care (ANC) plays an important role in maternal and neonatal health. By providing health contacts with the mother at key points in the continuum of care, quality ANC greatly reduces the risk of maternal mortality through preventive and promotive care and early detection and treatment of pregnancy-related complications, improving the survival and health of newborns.[2–4] In 2002, the WHO introduced the focused

## INTRODUCTION

The United Nations Inter-Agency Group estimated that in 2020 the global maternal mortality ratio was 223 deaths per 100000 live births, and UNICEF reported an average

ANC (FANC) model consisting of at least four ANC visits during pregnancy.

The Ministry of Health and Population (MoHP) in Nepal followed the FANC model with at least four ANC visits at the 4th, 6th, 8th and 9th month of gestation when they conducted the Nepal Demographic and Health Survey (DHS) in 2016.[5] To improve the utilisation of ANC, the Nepal government started a national Safe Delivery Incentive Programme, or Aama Programme in Nepali.[6] This programme provides monetary incentives to women who completed at least four ANC visits as suggested by the MoHP and women who delivered at health facilities by skilled birth attendants.[6] However, studies have found that recipients of the incentives were disproportionally wealthy families that had more access to health services and policy information, and the programme had limited effect on ANC utilisation in rural areas.[7 8] The MoHP published the National Medical Standard for Maternal and Newborn Care in 2020, stating that Nepal now recommends the new WHO eight contacts of ANC approach.[9]

According to the 2022 Nepal DHS, the ANC service utilisation rate was 94% for at least one ANC visit among women aged 15–49 years who had a live or stillbirth within 2 years before the survey; 80% of women had four or more ANC visits during their latest pregnancy and 82% of women in rural regions had at least four ANC visits.[10]

Household surveys, such as DHS and the Multiple Indicator Cluster Survey Programme, have been primary data sources for national-level health statistics across the world and will continue to be a major tool for routine tracking of coverage and quality of care in developing countries. Nepal has a national household survey every 5 years to evaluate the national ANC coverage, and the frequency of ANC visits serves as an important indicator. However, the survey often takes place many years after a woman's pregnancy. It is unknown whether the woman can correctly recall the total number of ANC visits and provide accurate answers to the DHS question. Therefore, the validity of this question in such household surveys is unknown. Previous studies have investigated the validity of ANC coverage indicators like quality of care, nutritional interventions, nutrition counselling and iron-folic acid supplementation in the same Nepal cohort, but the validity of frequency of ANC visits has not been explored.[11–13] The objective of this study is to examine the validity of maternal report of total number of ANC visits and factors associated with the accuracy of maternal report.

## METHODS
### Study site
This longitudinal cohort study was conducted from December 2018 to November 2020 within the study area of the Nepal Nutrition Intervention Project Sarlahi (NNIPS) located in the rural Sarlahi district of Southern Nepal. Sarlahi is a part of the Madhesh province bordered to the West by the Bagmati River and to the South by the state of Bihar, India. Two municipalities (Haripur and

Kabilasi) were chosen based on the census data and experiences from the local study team. Sarlahi district has a female population of 379 973 and approximately 181 624 (47.8%) of these are between 15 and 49 years of age.[14] Previous studies in the NNIPS area showed that women in Sarlahi district had an estimated pregnancy-related mortality ratio of 529 deaths per 100 000 live births in the period 2001–2006, which was almost two times of the national average.[15] Nepal DHS does not report maternal mortality ratios at the district level, so there is no more recent comparable data. Approximately 60% of women in this area attended four or more ANC visits in the period 2010–2016, which is lower than the average among rural regions.[11 15] Most ANC, especially in rural areas, is provided through public facilities, although there are some private facilities and hospitals. Five public health posts at Pharadwa, Laxmipur, Pidari, Pipariya and Kabilasi village development committees (VDCs) were designated to be the study sites because of their high attendance at ANC and accessibility to both the clients and the study team. VDCs have now been dissolved, but at the time of the study, VDCs were the smallest administrative unit in district where each VDC had nine wards.

### Study population, design and data collection
All pregnant women aged 15 years and older who lived in the NNIPS area and came for their first ANC visit, regardless of gestational age at this visit, to one of the five study health posts were eligible for the study. Women in the study were assumed to be married since it would be culturally inappropriate to ask about their marital status if they were pregnant and seeking ANC. Women who were younger than 15 years old were not enrolled. Women were considered ineligible to participate if they had already attended ANC or an ultrasound appointment before recruitment because not all ANC visits would be observed by the study team. Those who planned on visiting other health facilities than the five study ones for ANC during pregnancy were also considered ineligible for the same reason. Women who planned on leaving the NNIPS area during the study period, or up until 6 months after delivery, were excluded to prevent and minimise any loss to follow-up. Participants were consented at the enrolment visit and during the postpartum interview, respectively. All women signed consent with a witness signature for those who were illiterate. Married women aged 15–17 living with their husbands are considered emancipated minors in Nepal and the local institutional review board approved that they could consent for themselves.

The overall study approach is to assess the validity of maternal report by comparing the observed number of ANC visits (gold standard) to the answers provided by women in the 6-month postpartum interview. Trained field workers were present all day during regular hours (10:00 to 16:00) at the health posts. This was done to be able to observe all participant return visits for ANC to create the gold standard against which to compare maternal recall of number of visits. During the enrolment

period, trained field workers collected the demographic data of eligible participants, such as women's age, gestational age, parity and education level. Once enrolled, the participants were asked to complete a follow-up survey at each of their ANC visits. Trained field workers recorded their presence at the ANC visit and asked them questions about any health-seeking behaviour since the last visit. The follow-up form asked questions like 'what is the location of your most recent ANC visit' to help determine if the woman attended any ANC that was not observed by the study team. These direct observations served as the 'gold standard' for the validation analysis. A postpartum interview was conducted approximately 6 months after the woman's delivery to collect information on the ANC services they received during pregnancy. Some of the interview questions were constructed using the same language as the 2016 Nepal DHS. Specifically, the question about the number of ANC visits attended in the most recent pregnancy was identical to the question in the Nepal DHS ('How many times did you receive antenatal care during this pregnancy?'). The exact Nepali used in the Nepal DHS was used for this question. The interview also collected information on their socioeconomic status (SES) through questions about housing, household asset ownership, cooking fuels and ownership of land and household goods.

## Analysis

The study aimed to enrol 450 women to reach a sample size of 300, to estimate validation measures with sufficient precision (with prevalence of 50%, a 95% CI would be 13% wide or ±6.5% points), accounting for women who did not have a live birth, those who may have gone elsewhere for some ANC visits and did not have all visits observed and loss to follow-up. Eventually, 441 women were enrolled in the study and 434 of them participated in the postpartum interview.

The gold standard of observed number of visits was compared with the maternal report of the number of ANC visits for the validity analysis. Since it was impractical to follow women everywhere throughout their pregnancy, the follow-up survey at each ANC visit collected information to determine whether women received ANC at facilities other than the five designated health posts where observers were stationed. Participants were categorised into those who sought ANC elsewhere and those for whom all ANC was observed by the study team. In this way, a stricter gold standard was available for subgroup analysis.

The study cohort was categorised by the total number of ANC visits: 1–3 vs 4 or more (4+) visits; 1–3 visits, 4 visits, 5–6 visits and more than 6 visits. Since the Nepal MoHP recommended four or more ANC visits during pregnancy at the time of the study, the 4+ visits group was designed to see the compliance of FANC model and test the validity of a binary ANC frequency indicator. Individual validity was evaluated through sensitivity, specificity and area under the receiver operating characteristic curve

(AUC). To calculate sensitivity and specificity, 2×2 tables were constructed. Each participant was assigned to a cell in the table based on whether their ANC visit number fell in the group according to the gold standard and the maternal report. The calculation of sensitivity and specificity is similar to that of a diagnostic test. AUC in this scenario represents the probability that a woman's report of number of ANC visits is consistent with the gold standard category. AUC is calculated as the area under the plot of sensitivity versus (1−specificity).[16] An AUC higher than 0.7 is considered as high individual-level accuracy; an AUC of 0.5 indicates that maternal report on the indicator is no better than a random guess.[16] Population-level validity was measured through the inflation factor (IF), which gives an estimate of the accuracy of the postpartum survey in reflecting the true coverage in the population. It is calculated as the study coverage measured from maternal report divided by the true population coverage value based on the gold standard. The study coverage can be calculated using the formula: $Pr=P(SN+SP−1)+(1−SP)$, where Pr is the study coverage, P is the true population coverage, SN is sensitivity and SP is specificity.[16] An IF of 1.00 indicates perfect accuracy and an IF between 0.75 and 1.25 means there is low population-level bias.[16]

Bivariate and multivariate log-binomial regression models were used to assess factors associated with accuracy of maternal report. The primary outcome, accuracy, is a dichotomous variable. Maternal report of the number of ANC visits either matched with the categorical number of ANC visits observed (the gold standard), indicating accuracy, or it did not match (not accurate). Relative risk of accurately reporting was calculated because accurate reports were not rare outcomes; 38% of women recalled the number of ANC visits accurately according to the categorical definition described previously (1–3 vs 4+ visits; or 1–3 visits, 4 visits, 5–6 visits and more than 6 visits). Covariates related to maternal characteristics included maternal age, maternal education, number of prior live births and household SES. All covariates were included in the adjusted model. Maternal age was dichotomised into younger or older than 25 years. Any education was compared with no education and any previous live birth was compared with no previous live birth. The household SES variable was constructed based on family-owned land, animals and household items and housing infrastructures like types of cooking fuels, toilet and water sources. Housing characteristics were assigned scores and summed up for each woman. The total score was divided by the number of non-missing variables and separated into quartiles. Time between the postpartum interview and the last ANC observation was dichotomised to more or less than 1 year after examining its locally weighted scatterplot smoother (LOWESS) versus report accuracy. The intention was to interview all women at around 6 months postpartum. In practice, we did not know when they would deliver, so scheduled their postpartum visit 12 months after their first ANC visit if this was in the first or second trimester. If the first ANC visit was in the third

trimester, we scheduled the postpartum visit 6 months after the first ANC visit. The time between the last ANC visit and the postpartum visit would be somewhat longer than 6 months, since the last ANC visit could occur several months before birth. The observed total number of ANC visits was classified as 1–3, 4–7 and 8 or more using the LOWESS curve. Both LOWESS curves appeared linear in segments with a knot at approximately 1 year and knots at the fourth and eighth visits. A p value less than 0.05 was considered statistically significant.

All analyses were conducted using Stata V.17.0 (StatCorp).

### Patient and public involvement

Study participants were not involved in the design, recruitment, conduct or dissemination of this research. The 28-item checklist used for direct observation of the first and all subsequent ANC visits was reviewed by a local community advisory board in Nepal before the start of the study, but the public had no other part in the development or implementation of this study. There are no plans to disseminate results to the participants or community, aside from the local study staff who reside in the community.

### RESULTS

Among the 441 women enrolled in the study, 7 were lost to follow-up due to migration out of the study area and were not available for the postpartum interview. There was no difference between the background characteristics of the participants who were lost to follow-up and those who stayed in the study. Thirty-two women were excluded from the validation analysis because of their birth outcomes (not a live birth). At the time of the study, in the DHS, women with a pregnancy not resulting in a live birth were not asked the question about number of ANC visits (although more recent DHS do). In total, 402 women met the Nepal DHS sampling criteria and were included in the analysis. Among the 402 women, 228 reported receiving ANC at least once from non-study facilities, leaving 174 women with complete ANC observation by the study team. Figure 1 shows the flowchart of participants.

Table 1 summarises the maternal characteristics of women who attended the postpartum interview and had a live birth outcome. The age of women ranged from 16 to 41 years, with a mean age of 22.5 years. There was no significant age difference between women who sought ANC elsewhere and those who did not. The observed total number of ANC visits ranged from 1 to 14. On average, women attended 4.7 ANC visits during their pregnancy. The number of ANC visit was higher among women who sought ANC in non-study facilities. About 65% of women attended four or more ANC visits, the majority of which (72.7%) were women who reported receiving ANC from non-study clinics between observations at the study clinics. About 60% of women had not received any education.

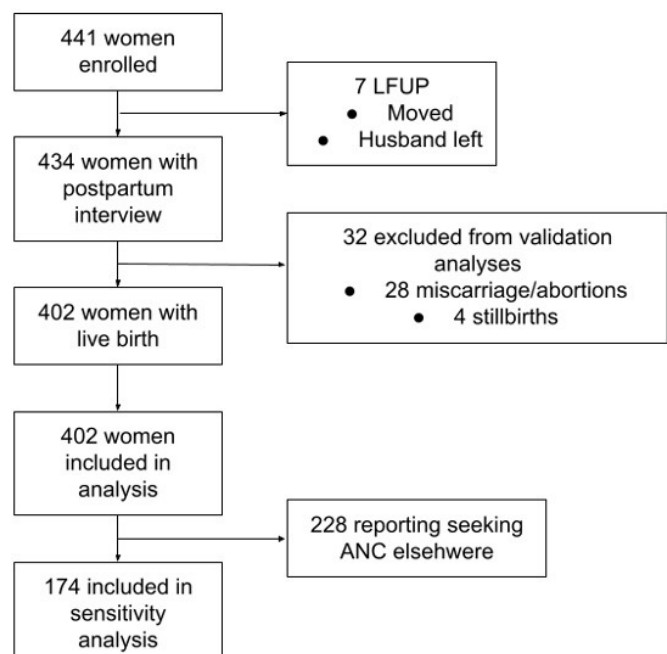

**Figure 1** Participant flowchart. ANC, antenatal care; LFUP, lost to follow-up.

Women who received ANC from non-study clinics were more educated and had higher SES.

Bland-Altman plots were constructed to compare ANC visit frequencies as observed by the gold standard and reported by maternal recall (online supplemental figure 1). In the entire cohort, the mean number of ANC visits observed was 4.7 (SD=2.5), compared with the reported number of 4.4 visits (SD=1.6). We observed both over-reporting and under-reporting of number of ANC visits, relative to the number observed (online supplemental figure 1A). Over-reporting was common among women who had fewer ANC visits, while under-reporting was common among higher ANC frequencies. In the subgroup of women whose ANC was fully observed (online supplemental figure 1B), the observed mean of total visits was 3.4 (SD=2.1), while the reported mean was 4.0 (SD=1.7). The distribution of observed total number of visits was positively skewed, with a long tail of women receiving 8+ visits, while the reported visits were more normally distributed (online supplemental figures 2 and 3). The disparity between the observed and reported distributions implied that women who had less or more than four ANC visits tended to report that they had four visits during pregnancy.

A total of 402 women with live births were included in the validity analysis, as per the Nepal DHS protocol. The validation results from the 402 women are shown in table 2A. The binary indicator of 4+ visits, which is used for global reporting and tracking, had a sensitivity of 89.2% (95% CI: 84.8% to 92.7%) and a specificity of 49.3% (95% CI: 40.8% to 57.8%). It showed a moderate level of individual validity (AUC: 0.69; 95% CI:0.65 to 0.74) and low population-level bias (IF: 1.17). The categorised visit groups, on the other hand, demonstrated poorer validity

**Table 1** Characteristics of participants with live births

| Characteristic | Observed all ANC visits (n=174) | | Received ANC between observations (n=228) | | Two sample t-test p value | Total (n=402) | |
| --- | --- | --- | --- | --- | --- | --- | --- |
| | Mean (SD) | Range | Mean (SD) | Range | | Mean (SD) | Range |
| Woman's age, years | 22.7 (4.4) | 16–41 | 22.3 (4.1) | 16–35 | 0.318 | 22.5 (4.2) | 16–41 |
| Total number of ANC visits observed | 3.4 (2.1) | 1–10 | 5.6 (2.3) | 2–14 | <0.01 | 4.7 (2.5) | 1–14 |
| Number of months between last ANC observation and postpartum interview | 11.2 (3.2) | 3–21 | 9.1 (2.5) | 3–17 | <0.01 | 10.0 (3.0) | 3–21 |

| | Observed all ANC visits (n=174) n (%) | Received ANC between observations (n=228) n (%) | $\chi^2$ p value | Total (n=402) n (%) |
| --- | --- | --- | --- | --- |
| 4 quantiles of SES | | | | |
| 1 | 76 (43.7) | 72 (31.6) | <0.01 | 148 (36.8) |
| 2 | 36 (20.7) | 35 (15.4) | | 71 (17.7) |
| 3 | 43 (24.7) | 83 (36.4) | | 126 (31.3) |
| 4 | 19 (10.9) | 38 (16.7) | | 57 (14.2) |
| Is this the woman's first pregnancy? | | | | |
| No | 133 (76.4) | 143 (62.7) | <0.01 | 276 (68.7) |
| Yes | 41 (23.6) | 85 (37.3) | | 126 (31.3) |
| Did the woman receive any years of education? | | | | |
| No | 121 (69.5) | 119 (52.2) | <0.01 | 240 (59.7) |
| Yes | 53 (30.5) | 109 (47.8) | | 162 (40.3) |
| Trimester at enrolment | | | | |
| 1–3 months | 63 (36.2) | 107 (46.9) | 0.043 | 170 (42.3) |
| 4–6 months | 106 (60.9) | 119 (52.2) | | 225 (56.0) |
| 7–9 months | 5 (2.9) | 2 (0.9) | | 7 (1.7) |
| Frequency of ANC visits | | | | |
| 1–3 visits | 103 (59.2) | 39 (17.1) | <0.01 | 142 (35.3) |
| 4 visits | 25 (14.4) | 48 (21.1) | | 73 (18.2) |
| 5–6 visits | 31 (17.8) | 71 (31.1) | | 102 (25.4) |
| More than 6 visits | 15 (8.6) | 70 (30.7) | | 85 (21.1) |

ANC, antenatal care; SES, socioeconomic status.

than the binary indicator in terms of sensitivity, AUC and IF. In general, sensitivity was low and had a declining trend with more ANC visits. The 1–3 visits category had the highest sensitivity score of 49.3% (95% CI: 40.8% to 57.8%). Specificity ranged from 63.5% (95% CI: 58.1% to 68.7%) in the 4 visits group to 94.0% (95% CI: 90.8% to 96.4%) in the more than 6 visits group. Only the 1–3 visits group showed a moderate level of individual validity (AUC: 0.69; 95% CI: 0.65 to 0.74), while other groups all had AUC less than 0.6 but barely better than a random guess. Population-level bias was common in all groups except the 5–6 visits group (IF: 1.10). There was a high overestimation of ANC visit frequency in the four visits group (IF: 2.12) and underestimation in the other two groups. However, the specificity of ANC categories was much better than that of the binary indicator. Specificity

ranged from 63.5% (95% CI: 58.1% to 68.7%) in the 4 visits group to 94.0% (95% CI: 90.8% to 96.4%) in the more than 6 visits group.

When considering only the subgroup with complete observation (table 2B), the binary indicator 4+ visits still had better sensitivity and AUC than the multicategorical variable. However, the IF increased to 1.51, indicating overestimation of four or more ANC visits at the population level. Sensitivity remained low among all visit categories and had the same decreasing trend in the overall population. Specificity was relatively similar. Individual validity was still highest but not very good in the 1–3 visits category (AUC: 0.66; 95% CI: 0.59 to 0.73) while others were no better than a random guess. However, two groups demonstrated great population-level validity. The 5–6 and more than six visits groups now had IFs of 1.10 and 1.00

**Table 2** Validation of maternal report of ANC visits

**(A) Among women with live births (n=402)**

| Gold standard versus reported | Sensitivity (95% CI), % | Specificity (95% CI), % | AUC (95% CI) | 'True' coverage (95% CI), % | Estimated survey coverage, % | Inflation factor |
|---|---|---|---|---|---|---|
| Number of ANC visits | | | | | | |
| FANC model (4 or more) | 89.2 (84.8 to 92.7) | 49.3 (40.8 to 57.8) | 0.69 (0.65 to 0.74) | 64.7 (59.8 to 69.4) | 75.6 | 1.17 |
| 1–3 | 49.3 (40.8 to 57.8) | 89.2 (84.8 to 92.7) | 0.69 (0.65 to 0.74) | 35.3 (30.6 to 40.2) | 24.4 | 0.69 |
| 4 | 47.9 (36.1 to 60.0) | 63.5 (58.1 to 68.7) | 0.56 (0.49 to 0.62) | 18.2 (14.5 to 22.3) | 38.6 | 2.12 |
| 5–6 | 30.4 (21.7 to 40.3) | 73.0 (67.6 to 77.9) | 0.52 (0.47 to 0.57) | 25.4 (21.2 to 29.9) | 27.9 | 1.10 |
| More than 6 | 21.2 (13.1 to 31.4) | 94.0 (90.8 to 96.4) | 0.58 (0.53 to 0.62) | 21.1 (17.3 to 25.5) | 9.2 | 0.44 |

**(B) Among women with live births and fully observed (n=174)**

| Gold standard versus reported | Sensitivity (95% CI), % | Specificity (95% CI), % | AUC (95% CI) | 'True' coverage (95% CI), % | Estimated survey coverage, % | Inflation factor |
|---|---|---|---|---|---|---|
| Number of ANC visits | | | | | | |
| FANC model (4 or more) | 80.3 (69.1 to 88.8) | 51.5 (41.1 to 61.4) | 0.66 (0.59 to 0.73) | 40.8 (33.4 to 48.5) | 61.5 | 1.51 |
| 1–3 | 51.5 (41.4 to 61.4) | 80.3 (69.1 to 88.8) | 0.66 (0.59 to 0.73) | 59.2 (51.5 to 66.6) | 38.5 | 0.65 |
| 4 | 40.0 (21.1 to 61.3) | 14.4 (9.5 to 20.5) | 0.54 (0.43 to 0.64) | 14.4 (9.5 to 20.5) | 33.3 | 2.32 |
| 5–6 | 19.4 (7.5 to 37.5) | 80.4 (73.0 to 86.6) | 0.50 (0.42 to 0.58) | 17.8 (12.4 to 24.3) | 19.5 | 1.10 |
| More than 6 | 20.0 (4.1 to 48.1) | 92.5 (87.2 to 96.0) | 0.56 (0.46 to 0.67) | 8.6 (4.9 to 13.8) | 8.6 | 1.00 |

ANC, antenatal care; FANC, focused antenatal care.

respectively, indicating low population-level bias. Overestimation still existed in the 4 visits group with an IF of 2.32.

Figure 2 is an IF graph created based on the sensitivity, specificity and true population coverage of the binary indicator (4+ visits) among women with live births. The difference between the observed and reported coverage is illustrated by the vertical red line. As outlined in the graph, maternal report tends to overestimate the number of ANC visits at lower numbers of visits, but underestimates at higher numbers of visits. Even in subgroups with IF close to 1.00, the survey estimation could greatly deviate from the true measurement depending on the true coverage of ANC4+.

Among the 402 women with live births, only 85 (21.1%) women's report matched exactly the ANC number as observed by the gold standard. Using categorical accuracy, 154 (38.3%) women reported correctly. The categorical accuracy rate was slightly higher in those who did not seek ANC elsewhere (41.4%) compared with those

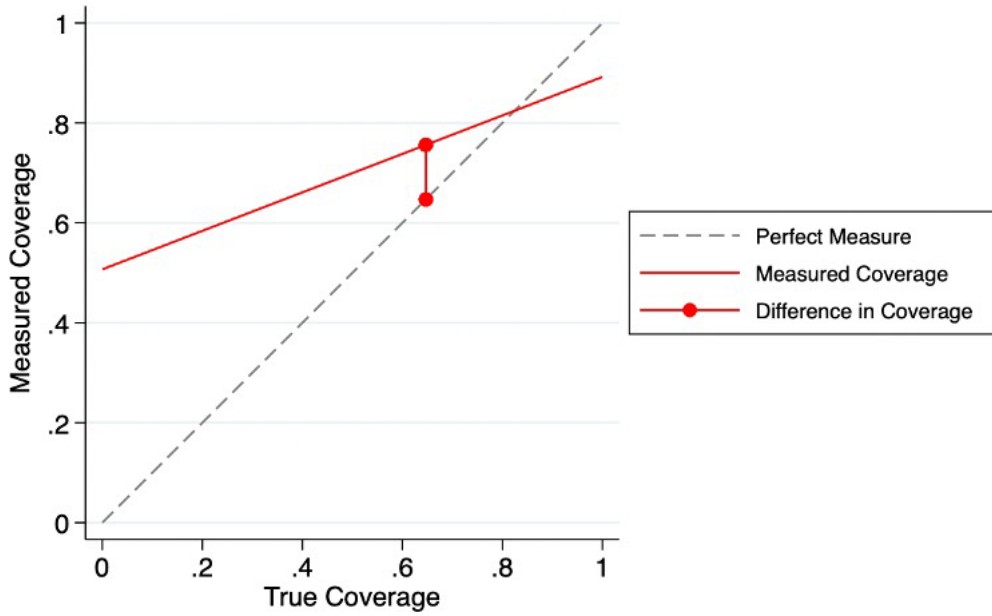

**Figure 2** Ture coverage compared with measured coverage for four or more ANC visits.

**Table 3** Maternal characteristics associated with report accuracy

| | n (%) | Unadjusted RR (95% CI) | Adjusted RR (95% CI) |
|---|---|---|---|
| Any education | 162 (40.3) | 0.97 (0.75 to 1.25) | 0.96 (0.73 to 1.24) |
| Any previous live birth | 126 (31.3) | 1.18 (0.92 to 1.53) | 1.22 (0.93 to 1.61) |
| Age≥25 | 122 (30.4) | 0.89 (0.67 to 1.18) | 0.92 (0.68 to 1.26) |
| SES quartiles (ref: first) | | | |
| 2 | 71 (17.6) | 0.93 (0.65 to 1.35) | 0.93 (0.64 to 1.34) |
| 3 | 126 (31.3) | 1.05 (0.79 to 1.41) | 0.98 (0.72 to 1.31) |
| 4 | 57 (14.2) | 0.81 (0.52 to 1.24) | 0.77 (0.50 to 1.21) |
| >1 year since last ANC observation | 76 (18.9%) | 1.17 (0.87 to 1.57) | 0.91 (0.67 to 1.24) |
| Number of ANC visits (ref: 1–3) | | | |
| 4–7 | 200 (49.8) | 0.70 (0.54 to 0.90)* | 0.66 (0.51 to 0.87)* |
| 8 or more | 60 (14.9) | 0.51 (0.32 to 0.81)* | 0.48 (0.30 to 0.77)* |

*P<0.05.
ANC, antenatal care; SES, socioeconomic status.

who visited other facilities (36.0%), but the difference was not statistically significant. Maternal characteristics such as education, previous birth, age and SES were not associated with reporting accuracy (table 3). The number of total ANC visits had the strongest association with maternal report accuracy, with increasing number of ANC visits associated with lower reporting accuracy. The unadjusted risk for women who received 4–7 and 8 or more ANC visits, compared with the 1–3 visit group, was 0.70 (95% CI: 0.54 to 0.90) and 0.51 (95% CI: 0.32 to 0.81), respectively. After adjusting for other variables, both RR decreased slightly to 0.66 (95% CI: 0.51 to 0.87) and 0.48 (95% CI: 0.30 to 0.77) and remained significant. This suggested that women with 4–7 ANC visits were 34% less likely to report this information correctly during household surveys, and women who had 8 or more ANC visits were 52% less likely to recall correctly, comparing to those attended 1–3 ANC visits.

## DISCUSSION

This study examined the validity of maternal report of the total number of ANC visits during pregnancy in rural Nepal using data from direct ANC observation. To our knowledge, it is the first study that validates number of ANC visits as an indicator in ANC coverage measurement. In general, individual-level validity was poor among women with four or more ANC visits and moderate among women with fewer ANC visits when using categorised ANC visits as an indicator, but was higher for the binary ANC4+ indicator. The validation results of the multicategorial variable showed that four ANC visits were often over-reported. Population-level bias seemed to be low among women with a higher number of ANC visits, but the survey question greatly overestimates the true coverage at lower prevalence and underestimates it at higher prevalence. Less than half of women recalled the exact number of

visits correctly during the postpartum visit. Reporting accuracy was found to be negatively associated with the total number of ANC visits during pregnancy but was not associated with maternal characteristics such as age, education, parity and SES. The recall period was also not associated with accuracy of recall but there was not a wide range of recall times to examine this variable.

These validation results suggest some bias in household surveys that report number of ANC visits and that report ANC4+. At the population level, 1–3 visits were under-reported, but having had four ANC visits was highly over-reported among the multicategorical variable. This might be due to Nepal's guideline on ANC, which was based on the FANC model, that may have introduced bias in household surveys with women more likely to report the norm or expected number of visits for which they would be paid through the cash incentive system. Only the 1–3 visits groups showed a moderate level of individual validity, which was consistent with the regression results where more ANC visits was associated with less accurate self-report. Besides the participant's ability to recall correctly, cognitive and situational issues are usually the two factors associated with self-report validity.[17] In this case, the language used during the postpartum interview, which was specifically designed to resemble that used in the DHS, could be misunderstood. A study of cognitive testing of questions about ANC suggested that over-reporting and under-reporting may be related to the definition of an ANC visit.[18] ANC visits are meant to be regular preventive checkups in pregnancy. However, if a woman came for care because she was sick, this would be counted as an ANC visit in the gold standard observed count but might not be counted as an ANC check-up visit by the woman at the time of recall 6 months postpartum.[18] Social desirability bias is the inclination of people to report more socially desired activities than they actually performed

(over-reporting) or understate undesirable attributes (under-report).[19] In the study scenario, women who had less than four ANC tended to report more and meet the social standard in front of the interviewer and sometimes the presence of their husbands, resulting in the under-reporting of 1–3 visits group and over-reporting of the 4 visits group. The low population-level bias here may be explained by the low prevalence of 5–6 visits due to a low number of false negatives. People who had more than six ANC visits seemed to under-report their receipt of care. This might be attributed to the respondents' inability to recall higher number of visits. ANC visits are often concentrated towards the end of pregnancy. Women might have conflated the visits in their minds and recalled a lower number. A study of social desirability bias was undertaken as part of this validity study. It showed very little social desirability bias but did show situational bias associated with whether family members or others were present during the postpartum interview.[20] It was found that the presence of any adult at the interview is associated with greater risk of overestimation of ANC frequency, with the presence of the husband being the most influential.[20]

There have been several yet limited studies on the validity of health indicators in coverage measurement. This paper contributes to the current body of validation studies and factors related to the accuracy of ANC self-report. One similar study evaluated the coverage rate of intermittent preventive treatment during pregnancy based on mother's recall in Benin, Ghana, Malawi and Tanzania.[21] It was found that compared with ANC card data (the gold standard), recalled data in household surveys were valid.[21] Sensitivity and specificity of self-report were generally higher than that in our study, and notably, the AUC of reported measurements from all four countries was higher than 0.8.[21] One potential reason for the different conclusions between the two studies could be that the recommended frequency for intermittent preventive treatment (at least three times) is lower than that of ANC, resulting in less variation in the total number of ANC visits and making it easier to recall correctly during household surveys. Additionally, ANC cards were used as the gold standard in their case instead of health facility records.[21] This could bias the results as women's self-reported validity might be associated with their ability to keep health records, which makes ANC cards not an optimal source for verification. Those who had a card would be more likely to read the card and be reminded of the number of visits they had.

In this study, the binary indicator 4+ visits performed better than multicategorical indicators (1–3 visits, 4 visits, 5–6 visits and more than 6 visits) in terms of both individual-level and population-level validity. In previous studies, dichotomous indicators often possessed higher validity than counts or frequency for the same intervention in household surveys. For example, the NNIPS study on iron-folic acid found that report of 'any iron-folic acid receipt' demonstrated better individual validity and very low population-level bias compared with specific tablet counts.[13] However, in that study, the prevalence of receipt of any iron-folic acid was very high (over 95%), which is likely the primary reason for higher validity and low bias. Furthermore, in a study comparing national household survey and health facility service statistics in Uganda, there was considerable agreement between the two data sources for skilled attendance at birth and at least four ANC visits.[22] However, if the number of ANC visits were dichotomised at eight times, the validity might not be better than that of categorical indicators. Many studies also have found that report accuracy was associated with the length of recall period, where accuracy decreases with extended duration of recall,[23–25] but such relation was not seen in the NNIPS studies.

A strength of this study is that the gold standard used in validation was through direct observation by trained field workers. Study observers were all trained to reach a standard level of validity before working at the study sites, which makes it a more objective and reliable source for verification than secondary databases. A second strength is that the study had a reasonable length of recall period, not as long as DHS but longer than many other studies. Other validation studies use recall periods of less than 6 months or even exit interviews to validate maternal report. One of the main limitations is that the study only considered women who presented for their first ANC at public health posts. Women who never attend ANC or those who do not go to public facilities were not captured through the study, but they may have characteristics that influence the overall self-report validity. Another limitation is that the study was unable to observe women if they went to other facilities for ANC. Subgroup analysis with just those women with all their ANC visits observed was conducted for more rigorous validation results. However, these measures were dependent on the women's ability to recall and report their care-seeking behaviour at other clinics. Lastly, the study was limited to only five health posts across two municipalities. Thus, the study result may not be generalizable to all women in the 20 Sarlahi municipalities, or Nepal's rural population in general.

## CONCLUSION

The DHS surveys are used in many countries to track progress in provision of ANC services and quality of maternal and newborn care. While the number of ANC visits does not imply quality of care, it is an important first step. If women are unable to accurately recall the number of ANC visits attended, this measure of progress is not very useful and ways to measure number of visits should be reconsidered. In general, the number of ANC visits as asked during DHS or household surveys, was not accurately recalled, although ANC4+ (a major marker of ANC coverage progress) recalled better than if ANC was more finely categorised. For women with more ANC visits than the standard of four (for Nepal at the time of this study), women tended to under-report the number of ANC visits. With the change from four or more to eight or more ANC

visits as the standard, approaches to improving recall should be identified and implemented.

**Contributors** MKM and JK designed the study. TP, SKK, SLC, JK and EB contributed to the implementation of the study. XX conducted the analyses and wrote the first draft of the manuscript. JK submitted edits to the manuscript. MKM, JK, TP, SKK, EB and SCL all have read and approved the final version of the manuscript. JK acted as the guarantor.

**Funding** This research was funded by the Bill and Melinda Gates Foundation (grant number OPP1172551).

**Competing interests** None declared.

**Patient and public involvement** Patients and/or the public were not involved in the design, or conduct, or reporting, or dissemination plans of this research.

**Patient consent for publication** Not applicable.

**Ethics approval** This study involves human participants. The institutional review board of the Johns Hopkins Bloomberg School of Public Health and the Nepal Health Research Council approved the parent study (IRB No. 00008808). Participants gave informed consent to participate in the study before taking part.

**Provenance and peer review** Not commissioned; externally peer reviewed.

**Data availability statement** Data are available upon reasonable request.

**ORCID iDs**
Xinyu Xie http://orcid.org/0009-0005-0966-5951
Melinda K Munos http://orcid.org/0000-0002-1349-8026
Emily Bryce http://orcid.org/0000-0002-4823-1647

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
