## [Reviewer comments · BMJ Open]

ARTICLE DETAILS

TITLE (PROVISIONAL)	Validation of maternal recall of number of antenatal care visits attended in rural Southern Nepal: a longitudinal cohort study
AUTHORS	Xie, Xinyu; Munos, Melinda; Plama, Tsering; Bryce, Emily; Khattry, Subarna; LeClerq, Steven C.; Katz, Joanne

VERSION 1 – REVIEW

REVIEWER	Namanya Basinda Catholic University of Health and Allied Sciences, Community Medicine
REVIEW RETURNED	21-Sep-2023

GENERAL COMMENTS	1. The introduction tells the history from 4 ANC visits to 8 ANC visits, however, this is not clearly linked up. It is therefore difficult to follow if its in the interest of the author to review the 4 vs 8 visits.2. No information on consent for under 18 provided? How did the author recruit women below the age of 18?3. Was there any ethical clearance procedure sought to allow the study to be conducted?
--

REVIEWER	Mahima Venkateswaran Nasjonalt folkehelseinstitutt, Global Health Cluster, Division for Health Services
REVIEW RETURNED	04-Oct-2023

GENERAL COMMENTS	Thank you for the opportunity to review this manuscript titled "Validation of maternal recall of antenatal care visits in rural Nepal". The authors have addressed an important topic in global maternal and child health. The methods and analysis are appropriate although with a relatively small sample size. Here are my comments. Abstract: 1. By "population bias" do you mean selection bias? I am a bit confused by why you call the result of 1-3 visits being underreported and 4 ANC visits overreported, population/selection bias. Could you perhaps rewrite this sentence?2. I am not sure what "multi-categorized ANC visits" means. I suggest you remove the term and rather state what it is. Introduction: 3. Household surveys like the DHS or MICS are the main data source for measuring ANC coverage (and other health indicators) in most LMICs and not just Nepal. These are the data that are also used in a lot of research on ANC coverage and metrics. The introduction section will be strengthened if the authors stated and
--

	described this on the global level than only Nepal. Methods: 4. Could you add some info on private health facilities and other providers/facilities for ANC in the study area? 5. Did you have any eligibility criteria based on when/at what gestational age the woman had her first visit? 6. Were the observers at the health facilities all days of the week and throughout the working day? Some more information on this could be useful. 7. “Time between the postpartum interview and the last ANC observation was dichotomized to more or less than 1 year after examining its locally weighted scatterplot smoother (LOWESS) versus report accuracy” – from your description I had assumed that women were interviewed at 6 months postpartum. Is that not so? Results: 8. The 228 women that reported receiving ANC from non-study facilities – was this any ANC, even if it was just 1 visit? Discussion: 9. I miss a discussion of what this means for other LMICs and to the field of MCH metrics/monitoring in general. This, I believe, is an important contribution of this study and should be highlighted. Also see my comment to the introduction section.
--	--

VERSION 1 – AUTHOR RESPONSE

Response to Reviewer 1’s comments:

1. The introduction tells the history from 4 ANC visits to 8 ANC visits, however, this is not clearly linked up. It is therefore difficult to follow if it’s in the interest of the author to review the 4 vs 8 visits. The introduction has been revised to focus on the 4-ANC-visit (FANC) model, which was recommended by the Nepal government when the study was conducted (page 5).
2. No information on consent for under 18 provided? How did the author recruit women below the age of 18?
Information on informed consent for women under and over 18 years old was provided on page 8.
3. Was there any ethical clearance procedure sought to allow the study to be conducted?
The Institutional Review Board of the Johns Hopkins Bloomberg School of Public Health and the Nepal Health Research Council approved the parent study (page 11, Ethical Review).

Response to Reviewer 2’s comments:

Abstract:

1. By “population bias” do you mean selection bias? I am a bit confused by why you call the result of 1-3 visits being underreported and 4 ANC visits overreported, population/selection bias. Could you perhaps rewrite this sentence?
The term has been changed to “population-level bias”, as distinct from individual-level bias (page 2).
2. I am not sure what “multi-categorized ANC visits” means. I suggest you remove the term and rather state what it is.
The term has been changed to “categorical indicators” followed by the specific categories of total number of ANC visits (page 2).

Introduction:

3. Household surveys like the DHS or MICS are the main data source for measuring ANC coverage (and other health indicators) in most LMICs and not just Nepal. These are the data that are also used in a lot of research on ANC coverage and metrics. The introduction section will be strengthened if the authors stated and described this on the global level than only Nepal.
A global perspective was added to the last paragraph of the introduction section (page 6).

Methods:

4. Could you add some info on private health facilities and other providers/facilities for ANC in the study area?

Information on public and private health facilities was added to the study site section (page 7).

5. Did you have any eligibility criteria based on when/at what gestational age the woman had her first visit?

Gestational age was not included in the eligibility criteria. Clarification was provided on page 7.

6. Were the observers at the health facilities all days of the week and throughout the working day?

Some more information on this could be useful.

Working hours of the trained field workers were provided on page 8.

7. "Time between the postpartum interview and the last ANC observation was dichotomized to more or less than 1 year after examining its locally weighted scatterplot smoother (LOWESS) versus report accuracy" – from your description I had assumed that women were interviewed at 6 months postpartum. Is that not so?

For women attending their first ANC visit at < 6 months gestation, we scheduled the postpartum interview 12 months later. For women attending their first ANC visit at >= 6 months gestation, we scheduled the postpartum interview 9 months later. Thus, the final interview will take place about 6 months postpartum. The time between the last ANC visit and the postpartum visit would be somewhat longer than 6 months since the last ANC visit could occur several months before birth. Clarification was provided on page 10-11.

Results:

8. The 228 women that reported receiving ANC from non-study facilities – was this any ANC, even if it was just 1 visit?

The 228 women who reported receiving ANC from non-study facilities went at least once to other clinics for ANC (page 12).

Discussion:

9. I miss a discussion of what this means for other LMICs and to the field of MCH metrics/monitoring in general. This, I believe, is an important contribution of this study and should be highlighted. Also see my comment to the introduction section.

The importance of this study was discussed in the conclusion section (page 20).

VERSION 2 – REVIEW

REVIEWER	Mahima Venkateswaran Nasjonalt folkehelseinstitutt, Global Health Cluster, Division for Health Services
REVIEW RETURNED	18-Nov-2023
GENERAL COMMENTS	Thank you for addressing my comments. I do not have any more feedback.